# Gender disparities in Italian academic medicine: A cross-sectional study of clinicians in the 2024 stanford top 2% scientists database

**Alessandro De Cassai**[1,2*], **Annalisa Boscolo**[1,2], **Maria Bisi**[1], **Christelle Chedom**[1], **Mario De Bernardo**[3], **Sofia Gazzea**[1], **Irene Paiusco**[1], **Paolo Navalesi**[1,2]

**1** Department of Medicine (DIMED), University of Padua, Padua, Italy, **2** Institute of Anesthesia and Intensive Care, University Hospital of Padua, Padua, Italy, **3** School of Medicine, University of Padua, Padua, Italy

* alessandro.decassai@unipd.it

## Abstract

### Background

Women remain underrepresented in senior and influential research positions despite gradual improvements over recent decades.The aim of this study was to assess gender representation among highly cited Italian clinical scientists and to examine differences across academic fields, institutions, and geographic regions.

### Methods

We conducted a cross-sectional analysis of Italian clinical scientists included in the 2024 Stanford Top 2 Percent Scientists database. Gender was determined from given names, and demographic and academic information was obtained from national physician registries and publicly available curriculum vitae. Characteristics were compared by gender, and a logistic regression model was used to examine factors associated with female representation.

### Results

A total of 3389 clinical scientists were identified, of whom 824 were women. Women represented approximately one quarter of highly cited Italian clinical researchers. They were younger and had more recent years of first publication compared with men. No significant gender differences were observed in academic affiliation or geographic distribution. Representation varied across fields, with surgical specialties demonstrating the lowest proportions of women and toxicology the highest. However, in most disciplines, representation converged toward the overall average when sample sizes increased, indicating that academic fields alone contributed minimally to gender representation. In the multivariable model, only toxicology showed

**Data availability statement:** Data types: Database is available from the original source: Ioannidis, John P.A. (2025), "August 2025 data-update for "Updated science-wide author databases of standardized citation indicators"", Elsevier Data Repository, V8, doi: 10.17632/btchxktzyw.8. Additional deidentified data (gender, age) is available from the authors How to access data: Data can be requested from alessandro.decassai@unipd.it.

**Funding:** The author(s) received no specific funding for this work.

**Competing interests:** The authors have declared that no competing interests exist.

significantly greater female representation, while more recent year of first publication was independently associated with being a woman..

## Conclusions

Women remain underrepresented among highly cited Italian clinical scientists, despite evidence of gradual improvement in representation over time. The limited influence of academic fields suggests that broader structural and cultural factors likely contribute to these gaps. These findings highlight the need for targeted strategies to promote equal representation and support the career advancement of women in academic medicine.

## Introduction

Gender inequalities persist across many professional fields, and although gender equity in academic medicine has improved in recent decades, women remain underrepresented among senior researchers and highly cited scientific leaders [1–3]. Persistent gender gaps in science, technology, engineering, and medicine (STEM) have been documented globally, with women facing structural, cultural, and institutional barriers that limit access to leadership roles, research funding, and recognition for scholarly contributions [4]. In academic medicine specifically, gender disparities emerge early in career trajectories and tend to widen over time, resulting in fewer women attaining full professorships, principal investigator status, and positions associated with substantial academic influence [5,6].

Highly cited scientists occupy a particularly important place within the academic ecosystem. Their work shapes research priorities, attracts funding, and influences scientific norms, policy discussions, and clinical practice guidelines. Citation metrics are commonly used—albeit imperfectly—as proxies for scientific impact, visibility, and intellectual leadership. Because these metrics can affect promotion, resource allocation, and professional reputation, gender disparities in citation impact may reinforce or exacerbate pre-existing inequities [7]. Prior literature indicates that women often receive fewer citations than men even when controlling for publication volume and field, a phenomenon attributed to factors such as collaboration structures, authorship practices, network homophily, and biases in peer evaluation [8]. Yet, despite increasing attention to these issues, comprehensive and discipline-specific data on the gender distribution of highly cited medical researchers remain limited.

International reports, including the UNESCO Science Report and the European Commission's *She Figures*, highlight persistent underrepresentation of women in senior scientific positions, particularly in biomedical research environments that rely heavily on competitive funding and citation-based indicators of excellence [9,10]. At the national level, Italy mirrors many of these global patterns. Although the proportion of women in medical schools and early academic stages has increased, women remain less prevalent in high-ranking academic positions, leadership roles in scientific societies, and major grant awards [11]. Regional differences across Italy

may further contribute to heterogeneity in academic advancement, with institutional cultures, resources, and local policies influencing opportunities available to researchers.

The Stanford Top 2% Scientists database [12], which compiles standardized citation indicators across disciplines, provides an opportunity to examine gender disparities among the most influential scientists at a national scale. Of note,in this study we will refer to gender disparity as an observed imbalance in representation, without implying causation. Because the list includes living researchers active across a wide range of scientific domains, it offers a useful snapshot of scientific leadership, visibility, and scholarly output. However, analyses focusing specifically on medical disciplines—particularly within individual countries—are scarce.

This study examines gender disparities across medical fields among Italian scientists included in the 2024 Stanford Top 2% Scientists database. We aim to provide a detailed assessment of the distribution of women and men among Italy's most highly cited medical researchers in order to improve the understanding patterns of scientific influence and promote equity, diversity, and inclusion within academic medicine.

## Materials and methods

This study has been reported with the support of the STROBE checklist (See S1 Text, STROBE checklist). We conducted a cross-sectional analysis of Italian medical scientists listed in the *2024 Stanford Top 2% Scientists* database, an extensively used ranking that compiles standardized citation metrics across disciplines [12]. The database provides composite citation indicators based on Scopus data, including total citations, h-index, and field-weighted metrics, enabling the identification of researchers with high scientific impact. For the purposes of this study, we extracted all individuals classified under *clinical medicine*, which includes a broad range of medical and surgical specialties as defined by the Stanford classification system. Because the database provides affiliations and disciplinary categories but does not include demographic variables, additional steps were required to determine gender and age.

A significant limitation in the literature examining gender distributions among researchers is the challenge of accurately assigning gender based solely on given names. This issue is particularly acute in international datasets, where names may be culturally ambiguous, unisex, or transliterated inconsistently. To minimize misclassification bias, we restricted our analysis to Italian scientists, as Italian given names allow gender inference with high accuracy due to morphological regularities, although exceptions prevent deterministic classification. Moreover, the Italian National Institute of Statistics (ISTAT), the official governmental body responsible for collecting and analyzing demographic data in Italy, provides a publicly accessible database (https://www.istat.it/dati/calcolatori/contanomi/; last accessed 01 April 2026) listing Italian given names along with the corresponding percentages of individuals by gender. Gender was determined through a rule-based classification of given names, verified through independent manual checks when necessary. In cases where names could theoretically have variants across regions or cultures, additional verification was conducted using professional web pages or publicly available biographical information.

Age information was obtained primarily from the Federazione Nazionale Ordini dei Medici Chirurghi e Odontoiatri (FNOMCeO) registry, an official national database containing demographic and professional information for all licensed physicians in Italy [13]. When age or year of birth was not available through FNOMCeO, we consulted publicly accessible curriculum vitae or institutional profiles, ensuring consistency through cross-referencing when multiple sources were present. Year of first publication, used as a proxy for career onset, was extracted from citation profiles available through the Stanford dataset.

Geographic regional classification was based on the institutional affiliation listed in the Stanford dataset. Each university or research institution was assigned to one of the three traditional Italian macro-regions—North, Center, or South—following the standard Italian National Institute of Statistics (ISTAT) classification. For researchers with multiple affiliations, the primary academic institution was used.

Descriptive statistics were computed for demographic and professional characteristics, stratified by gender. Categorical variables (e.g., geographic region, field of specialization, university affiliation) were compared using χ² tests, while

continuous variables with non-normal distributions—such as total citations, h-index, and year of first publication—were analyzed using Wilcoxon rank-sum tests. Normality was assessed through visual inspection of density plots and Shapiro–Wilk tests.

To evaluate factors associated with female gender among highly cited Italian medical scientists, we constructed a multivariable logistic regression model. Predictor variables included university affiliation (categorized into major academic institutions versus all others), year of first publication, scientific field (grouped according to major clinical disciplines), and geographic macro-region. Age was excluded from the model due to multicollinearity with year of first publication, as both variables capture overlapping temporal aspects of the career stage. Variance inflation factors (VIFs) were assessed to confirm acceptable levels of collinearity among included predictors.

All statistical analyses were performed using R statistical software (version 4.4.2; R Foundation for Statistical Computing, Vienna). Standard packages for regression modeling, data manipulation, and visualization were employed. The threshold for statistical significance was set at $p < 0.05$ for all analyses. Full model outputs, sensitivity analyses, and field-specific $\chi^2$ tests are reported in the Supplementary Material to ensure transparency and reproducibility.

A Large Language Model (ChatGPT 5.2; OpenAI, San Francisco, CA, USA) was used solely to review and improve the grammar and fluency of the manuscript. All content was critically reviewed and verified by the authors, who retained full responsibility and accountability for the final version of the manuscript [14].

## Results

A total of 3389 Italian clinical scientists were identified in the 2024 Stanford Top 2% Scientists database and met the eligibility criteria. Of these, 824 (24.3%) were women, indicating that women constitute roughly one in four highly cited Italian medical researchers. Table 1 summarizes the demographic and academic characteristics stratified by gender.

Notably we were not able to retrieve the age data for 125 women and 169 men. Women were significantly younger than men (median age 55 vs 61 years; *p-value* < 0.001), and their careers appeared to have started later, as reflected in a more recent year of first publication (1997 vs 1992; *p-value* < 0.001). Women also exhibited marginally but significantly lower H-index and Hm-index values compared with men (both *p-value* < 0.001). These differences likely reflect, at least in part, age and cohort effects rather than inherent disparities in productivity.

No significant gender differences were observed in academic affiliation: 72.9% of women and 72.5% of men held university affiliations (*p-value* 0.570). Similarly, geographic distribution across Italy (North, Center, South) did not differ

**Table 1. Demographic and academic characteristics of Italian clinical scientists, stratified by gender.**

| Variable | Female (n: 824) | Male (n: 2565) | P value |
|---|---|---|---|
| **Scientific field** | See eTable1 | | <0.001 |
| **Academic affiliation** | 72.9% (601) | 72.5% (1859) | 0.570 |
| **Geographic zone** | | | |
| North Italy | 52.1% (429) | 53.8% (1380) | 0.626 |
| Center | 27.1% (223) | 26.8% (687) | |
| South of Italy | 20.7% (171) | 19.4% (498) | |
| **H-index** | 13 (11-16) | 14 (11-17) | <0.001 |
| **Hm-index** | 4.92 (3.99-5.92) | 4.98 (4.25-6.11) | <0.001 |
| **First article year** | 1997 (1988-2006) | 1992 (1983-2003) | <0.001 |
| **Age\*** | 55 (46-64) | 61 (49-69) | <0.001 |

Data are presented as n (%) or median (first–third quartile). *Age was unavailable for 125 women and 169 men.

by gender (*p-value* 0.626). These findings suggest that institutional type and regional location do not appear to drive the observed gender disparities among top-cited scientists.

Gender representation varied substantially across scientific fields (see S2 Text, eTable 1). Several disciplines demonstrated relatively balanced representation or even female predominance in small samples (e.g., Epidemiology, Gerontology, History of Science), although these fields had low numbers of Italian scientists accounted for in the analyzed database. Conversely, certain disciplines remained markedly male-dominated. Surgical specialties showed the lowest proportion of women (6.1%), followed by ophthalmology, orthopedics, otorhinolaryngology, and anesthesiology. At the opposite end of the spectrum, Toxicology exhibited the highest female representation (63.2%), followed by Microbiology and Nutrition & Dietetics.

In the multivariable logistic regression model evaluating factors associated with female gender (see S2 Text, eTable 2), only Toxicology demonstrated a statistically significant higher likelihood of women's representation (OR 7.66; 95% CI, 2.08 to 37.57; *p-value* 0.005). Most other disciplines showed wide confidence intervals and non-significant associations, largely due to small sample sizes in several fields. The overall pattern reflected heterogeneity at the extremes but a tendency toward the population average (~24%) as the denominator of scientists increased.

The funnel plot (Fig 1) further supported this interpretation: fields with a larger number of scientists had proportions of women clustering closely around the overall mean, whereas more extreme values—both high and low—occurred primarily in fields with very small sample sizes. This suggests that discipline alone is not a strong determinant of women's representation when sample size variability is accounted for

Among continuous predictors, year of first publication was independently associated with female gender (OR 1.02 per year; 95% CI, 1.02–1.03; $P<.001$). This indicates that researchers entering the scientific community more recently were

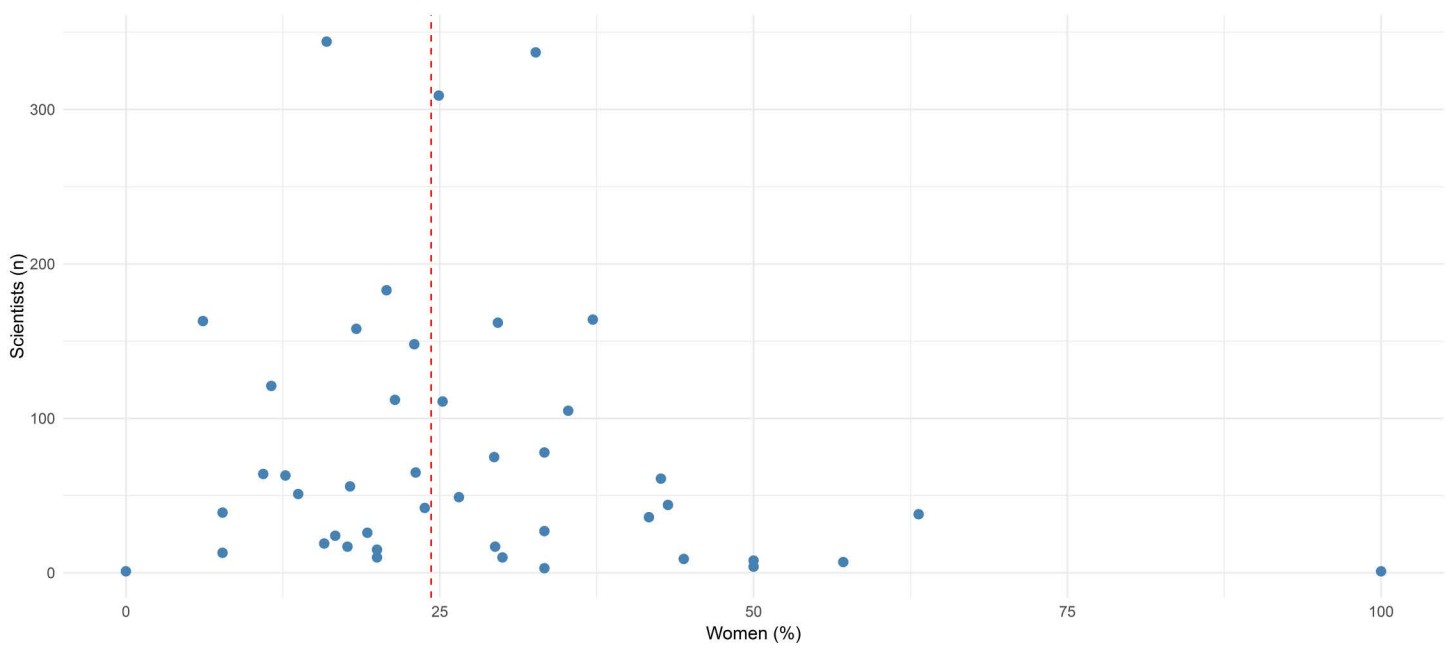

**Fig 1. Gender distribution among top Italian clinical scientists by discipline.** Data include 3389 Italian scientists in academic medical fields in the 2024 Stanford Top 2% Scientists database. The proportion of women (n: 824) is shown for each medical specialty. Funnel plot illustrates the relationship between the number of scientists in each discipline and female representation, indicating convergence toward the overall average (dashed line: 24.3% women). Surgical specialties had the lowest proportion of women (6.1%), whereas toxicology had the highest (63.2%).

increasingly likely to be women, consistent with a gradual improvement in gender representation over time. Age was excluded from the model due to multicollinearity with year of first publication. Importantly, assessment of variance inflation factors (VIFs) confirmed that multicollinearity among the included predictors remained at acceptable levels, supporting the stability and interpretability of the regression estimates.

Neither academic affiliation (OR 0.93; *p-value* 0.472) nor geographic macro-region (Center [reference], North [OR 0.93;95%CI 0.77 to 1.13, p-value 0.469], South [OR 0.93; 95%CI 0.73 to 1.19; p-value 0.585] demonstrated significant associations with gender in the multivariable model. Thus, after accounting for field and cohort, institutional and regional factors did not meaningfully influence the likelihood of being a woman among highly cited Italian medical scientists.

## Discussion

Overall, women's representation among highly cited Italian clinical scientists remains low, with only one in four scientists being women. This proportion aligns with longstanding global evidence showing that women remain underrepresented in senior academic medicine and among researchers with the highest citation impact or leadership visibility [2,15]. Although toxicology displayed a substantially higher proportion of women, the funnel plot demonstrated that, in most disciplines, gender representation converged toward the overall average, indicating that the disciplinary field alone exerts only a modest influence on gender disparities. This echoes findings from prior international studies showing that while gender distribution varies somewhat by specialty, structural and systemic factors—rather than disciplinary characteristics—tend to drive disparities [4,16].

The association between earlier year of first publication and male gender suggests that gender gaps in representation have gradually narrowed over successive academic cohorts. This pattern is consistent with broader European and global trends documented in the *She Figures* report [9] and the UNESCO Science Report [10], both of which describe steady increases in women's entry into biomedical research over recent decades. Nevertheless, the persistence of a sizable gender gap in the upper tiers of citation impact suggests the presence of the "leaky pipeline" phenomenon, wherein women enter academic medicine in increasing numbers but remain underrepresented in positions of greatest influence [2,17]. Although the cross-sectional design of the present study does not allow direct testing of the leaky pipeline hypothesis, the association between more recent career onset and female gender is consistent with generational shifts described in longitudinal analyses of academic medicine. Factors such as differential access to resources, cumulative advantage mechanisms, and citation biases may contribute to these underrepresentations [18].

Despite encouraging signs of generational improvement, gender differences in H-index and Hm-index remained evident. While these metrics are influenced by career length, research has shown that women often receive fewer citations than men even after adjusting for publication volume, collaboration networks, and field-specific norms [19]. Such patterns may reflect systemic biases in authorship practices, peer review, and citation behavior, including well-documented gender gaps in self-citation [7].

Beyond bibliometric and cohort effects, multiple structural and cultural determinants have been shown to influence gender representation in academic medicine. Prior research indicates that women face differential access to leadership opportunities, grant funding, and senior authorship positions, often shaped by implicit bias in evaluation processes and cumulative advantage mechanisms that favor established networks [15,16,18]. Studies examining peer-review and funding decisions suggest that assessments may be influenced not only by the scientific proposal but also by perceptions of the applicant, potentially contributing to slower career progression among women [15]. In addition, qualitative and survey-based research has documented the impact of hostile work environments, harassment, and gender-based discrimination within academic medicine, factors that may reduce retention and advancement of women over time [4,16]. Motherhood and caregiving responsibilities also represent important structural variables: evidence shows that parenthood disproportionately affects women's research productivity and promotion trajectories, particularly during early and mid-career stages when citation accumulation is most sensitive to output continuity [5,6]. These interacting

mechanisms—cultural expectations, implicit bias, structural inequities, and differential caregiving burdens—likely contribute to the persistent underrepresentation of women in positions associated with high citation impact, even in the absence of overt institutional exclusion.

A major strength of this study is the high accuracy of gender identification, enabled by the use of Italian given names, which are generally unambiguous. This reduces the risk of misclassification, a limitation that often affects international bibliometric research on gender. Additionally, by focusing on a comprehensive national sample of highly cited clinical researchers, this study provides a detailed and field-level assessment of representation within Italy's academic medicine landscape.

However, several limitations merit consideration. First, the Stanford Top 2% Scientists database relies on Scopus citation metrics, which may vary by field, subdiscipline, and citation culture. Our study could not account for career interruptions, which disproportionately affect women and may influence cumulative citations. Nor did we have data on mentorship, grant funding, authorship position, or institutional policies, all known to influence gender disparities in academic advancement. Additionally, although we examined variation across Italian macro-regions, the findings may not be generalizable to countries with different academic structures or policy environments. Our analyses are not designed to identify causal mechanisms underlying observed imbalances. Accordingly, the findings should not be interpreted as evidence of discrimination at the individual, institutional, or disciplinary level, nor do they allow assessment of specific structural drivers (e.g., hiring, promotion, funding, mentorship, caregiving-related career interruptions) that may shape career trajectories and citation behaviours. Finally, our study does not evaluate the effectiveness, appropriateness, or impact of equity- or inclusion-oriented policies or programs; rather, it documents patterns of representation within a citation-defined cohort to inform future hypothesis-driven and longitudinal research.

These results underscore the need for targeted strategies to support women's career advancement in academic medicine. Addressing structural and cultural barriers—such as unequal caregiving expectations, biases in evaluation, and disparities in research resources—remains essential to ensure that women are equitably represented among the scientific leaders who shape clinical research and practice.

## Supporting information

**S1 Text. STROBE Checklist.**
(DOCX)

**S2 Text. eTable1 and eTable2.**
(DOCX)

## Author contributions

**Conceptualization:** Alessandro De Cassai, Annalisa Boscolo.

**Data curation:** Alessandro De Cassai, Mario De Bernardo, Sofia Gazzea, Irene Paiusco.

**Formal analysis:** Alessandro De Cassai, Maria Bisi.

**Investigation:** Alessandro De Cassai.

**Methodology:** Alessandro De Cassai.

**Project administration:** Alessandro De Cassai.

**Software:** Alessandro De Cassai.

**Supervision:** Alessandro De Cassai, Paolo Navalesi.

**Validation:** Alessandro De Cassai.

**Visualization:** Alessandro De Cassai.

**Writing – original draft:** Alessandro De Cassai, Annalisa Boscolo, Maria Bisi, Christelle Chedom, Mario De Bernardo, Sofia Gazzea, Irene Paiusco.

**Writing – review & editing:** Alessandro De Cassai, Annalisa Boscolo, Maria Bisi, Christelle Chedom, Mario De Bernardo, Sofia Gazzea, Irene Paiusco, Paolo Navalesi.

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
