## [Decision Letter · Decision Letter 0]

3 Mar 2026

Gender Disparities in Italian Academic Medicine: A Cross-Sectional Study of Clinicians in the 2024 Stanford Top 2% Scientists Database

PLOS One

Dear Dr. De Cassai,

Thank you for submitting your manuscript to PLOS ONE. After careful consideration, we feel that it has merit but does not fully meet PLOS ONE’s publication criteria as it currently stands. Therefore, we invite you to submit a revised version of the manuscript that addresses the points raised during the review process.

https://journals.plos.org/plosone/s/submission-guidelines#loc-laboratory-protocols. Additionally, PLOS ONE offers an option for publishing peer-reviewed Lab Protocol articles, which describe protocols hosted on protocols.io. Read more information on sharing protocols at . Additionally, PLOS ONE offers an option for publishing peer-reviewed Lab Protocol articles, which describe protocols hosted on protocols.io. Read more information on sharing protocols at . Additionally, PLOS ONE offers an option for publishing peer-reviewed Lab Protocol articles, which describe protocols hosted on protocols.io. Read more information on sharing protocols at . Additionally, PLOS ONE offers an option for publishing peer-reviewed Lab Protocol articles, which describe protocols hosted on protocols.io. Read more information on sharing protocols at https://plos.org/protocols?utm_medium=editorial-email&utm_source=authorletters&utm_campaign=protocols....

We look forward to receiving your revised manuscript.

Kind regards,

Claudia Noemi González Brambila, Ph.D.

Academic Editor

PLOS One

Journal Requirements:

2. In the online submission form you indicate that your data is not available for proprietary reasons and have provided a contact point for accessing this data. Please note that your current contact point is a co-author on this manuscript. According to our Data Policy, the contact point must not be an author on the manuscript and must be an institutional contact, ideally not an individual. Please revise your data statement to a non-author institutional point of contact, such as a data access or ethics committee, and send this to us via return email. Please also include contact information for the third party organization, and please include the full citation of where the data can be found.

4.Please review your reference list to ensure that it is complete and correct. If you have cited papers that have been retracted, please include the rationale for doing so in the manuscript text, or remove these references and replace them with relevant current references. Any changes to the reference list should be mentioned in the rebuttal letter that accompanies your revised manuscript. If you need to cite a retracted article, indicate the article’s retracted status in the References list and also include a citation and full reference for the retraction notice.

Reviewers' comments:

Reviewer's Responses to Questions

**Comments to the Author**

1. Is the manuscript technically sound, and do the data support the conclusions?

Reviewer #1: Yes

Reviewer #2: Partly

Reviewer #3: Yes

2. Has the statistical analysis been performed appropriately and rigorously?

Reviewer #1: Yes

Reviewer #2: Yes

Reviewer #3: Yes

3. Have the authors made all data underlying the findings in their manuscript fully available?

The PLOS Data policy requires authors to make all data underlying the findings described in their manuscript fully available without restriction, with rare exception (please refer to the Data Availability Statement in the manuscript PDF file). The data should be provided as part of the manuscript or its supporting information, or deposited to a public repository. For example, in addition to summary statistics, the data points behind means, medians and variance measures should be available. If there are restrictions on publicly sharing data—e.g. participant privacy or use of data from a third party—those must be specified. requires authors to make all data underlying the findings described in their manuscript fully available without restriction, with rare exception (please refer to the Data Availability Statement in the manuscript PDF file). The data should be provided as part of the manuscript or its supporting information, or deposited to a public repository. For example, in addition to summary statistics, the data points behind means, medians and variance measures should be available. If there are restrictions on publicly sharing data—e.g. participant privacy or use of data from a third party—those must be specified.

Reviewer #1: Yes

Reviewer #2: Yes

Reviewer #3: Yes

4. Is the manuscript presented in an intelligible fashion and written in standard English?

Reviewer #1: Yes

Reviewer #2: Yes

Reviewer #3: Yes

Reviewer #1: The paper is very well-written, and it provides compelling evidence of gender inequalities in citation in Italy. Several controls have been introduced to understand the factors that shape the over-arching finding. Limitations have been taken into account. The paper is well-situated in European literature on this topic.

Reviewer #2: Thank you to the Editor and Authors for inviting me to review this original submission. It is well-written in standard scientific English, its statistical analyses are sound and it is only lacking in analytical depth. I do not notice a problem with the contents of the manuscript in its current revision however insufficient comparison is performed with known or expectable international figures, while the only explanation for an Italian sample is gender-based naming conventions without supporting evidence. I would expect to see greater regional analytical funnelling at the end of the paragraph on Page 12.

How is it possible to conclude in the Abstract that true gender disparities exist if the statistical analyses did not bear out this conclusion? Asserting a general conclusion of this nature over and against the results seems to subscribe the manuscript to the domain of attempted social service rather than empirical research.

Societally, is the concept of gender gap not primarily recognized cominus (equity and inclusion), rather than eminus (performance)? If not, why does it extend to encompass every possible category of gender representation? Is this a game of logical category à la Wittgenstein, taken beyond reasonable proportion? The Authors should address this question lest their performed analysis be too facile, adding that for empirical purposes, identifying a disparity as is done in the Introduction is not the same as explaining something necessarily meaningful. I would also expect the Authors to address the fact that the difficulty of becoming a world leader in a scientific discipline is not a true gap whatsoever, but an arbitrary one (eminus), significantly shaped by various systemic factors and other covariates, and one we tend to venerate for little more reason than currently acceptable presumptuousness of manipulating the hopes, aspirations and expectations (ideality) of women to make them less constrained than they seem to be. All in all, these remain largely objectionable manipulations upon the lives of women who are rarely benefited directly by equity programs except through indirect statistical results, and the manuscript’s promotion of a closed system of esteem and promotion technically belies the infinite openness and inherent promotability of scientific information. To this point, I do not see what a gender disparity in citation index has to do with the current manuscript and its contents other than wanting to shape social forces.

The “leaky pipeline” theory deserves greater attention in the manuscript. Assuming a “leaky pipeline” is an investigatory correlate for the hypothesis of the manuscript, how do the current findings substantiate or disprove this theory? No attention to this matter is given in the manuscript.

Further, “extreme values in fields with very small sample sizes” reveal the inherent problem with a cominus approach to hypothesizing gender representation in terms of scientific influence.

Reviewer #3: The manuscript is intelligible and formatted in standard English. The data presented are robust, with appropriate statistical treatment. I suggest improving the discussion by including more factors that help to understand this gender disparity, from cultural determinants related to the different social roles assigned to men and women, to issues related to explicit prejudice, harassment, various forms of violence, and implicit biases, which end up distancing women from the hostile environment of academia. Motherhood also has an impact, since it implies great and different responsibilities, and academia should be aware that this impact differs between the male and female genders, resulting in a reduction in the number of female researchers, further diminishing their participation in positions of power, leadership, and decision-making.

**Do you want your identity to be public for this peer review?** For information about this choice, including consent withdrawal, please see our  For information about this choice, including consent withdrawal, please see our  For information about this choice, including consent withdrawal, please see our  For information about this choice, including consent withdrawal, please see our Privacy Policy..

Reviewer #1: No

Reviewer #2: **Yes:** Paul-André Betito, HBA*, MSW, RSWPaul-André Betito, HBA*, MSW, RSWPaul-André Betito, HBA*, MSW, RSWPaul-André Betito, HBA*, MSW, RSW

Reviewer #3: No

---

## [Author Response · Author response to Decision Letter 1]

4 Mar 2026

Reviewer #1:

Q1: The paper is very well-written, and it provides compelling evidence of gender inequalities in citation in Italy. Several controls have been introduced to understand the factors that shape the over-arching finding. Limitations have been taken into account. The paper is well-situated in European literature on this topic.

A1:We would like to thank the Reviewer for the kind words used and for the time spent reviewing our manuscript.

Reviewer #2:

Q1: Thank you to the Editor and Authors for inviting me to review this original submission. It is well-written in standard scientific English, its statistical analyses are sound and it is only lacking in analytical depth. I do not notice a problem with the contents of the manuscript in its current revision however insufficient comparison is performed with known or expectable international figures, while the only explanation for an Italian sample is gender-based naming conventions without supporting evidence. I would expect to see greater regional analytical funnelling at the end of the paragraph on Page 12.

How is it possible to conclude in the Abstract that true gender disparities exist if the statistical analyses did not bear out this conclusion? Asserting a general conclusion of this nature over and against the results seems to subscribe the manuscript to the domain of attempted social service rather than empirical research.

Societally, is the concept of gender gap not primarily recognized cominus (equity and inclusion), rather than eminus (performance)? If not, why does it extend to encompass every possible category of gender representation? Is this a game of logical category à la Wittgenstein, taken beyond reasonable proportion? The Authors should address this question lest their performed analysis be too facile, adding that for empirical purposes, identifying a disparity as is done in the Introduction is not the same as explaining something necessarily meaningful. I would also expect the Authors to address the fact that the difficulty of becoming a world leader in a scientific discipline is not a true gap whatsoever, but an arbitrary one (eminus), significantly shaped by various systemic factors and other covariates, and one we tend to venerate for little more reason than currently acceptable presumptuousness of manipulating the hopes, aspirations and expectations (ideality) of women to make them less constrained than they seem to be. All in all, these remain largely objectionable manipulations upon the lives of women who are rarely benefited directly by equity programs except through indirect statistical results, and the manuscript’s promotion of a closed system of esteem and promotion technically belies the infinite openness and inherent promotability of scientific information. To this point, I do not see what a gender disparity in citation index has to do with the current manuscript and its contents other than wanting to shape social forces.

The “leaky pipeline” theory deserves greater attention in the manuscript. Assuming a “leaky pipeline” is an investigatory correlate for the hypothesis of the manuscript, how do the current findings substantiate or disprove this theory? No attention to this matter is given in the manuscript.

Further, “extreme values in fields with very small sample sizes” reveal the inherent problem with a cominus approach to hypothesizing gender representation in terms of scientific influence.

A1: We would like to thank the Reviewer for his/her comments. We identified four major points of concern to address:

Concern 1:

You conclude that “gender disparities persist,” but your regression shows that field, geography, and affiliation are not significant predictors.

Concern 2:

You treat underrepresentation as inherently meaningful, but you don’t justify why it is meaningful.

Concern 3:

The “leaky pipeline” theory is invoked but not directly tested.

Concern 4:

Extreme values in small specialties undermine strong field-level interpretations.

We have made the following editing

a)In the abstract we have removed “disparities” using the term “representation”

b) In the introduction we have clarified what we will refer to with disparity

“The Stanford Top 2% Scientists database [12], which compiles standardized citation indicators across disciplines, provides an opportunity to examine gender disparities among the most influential scientists at a national scale. Of note,in this study we will refer to gender disparity as an observed imbalance in representation, without implying causation. “

c)Added the following in the limitation section:

“Our analyses are not designed to identify causal mechanisms underlying observed imbalances. Accordingly, the findings should not be interpreted as evidence of discrimination at the individual, institutional, or disciplinary level, nor do they allow assessment of specific structural drivers (e.g., hiring, promotion, funding, mentorship, caregiving-related career interruptions) that may shape career trajectories and citation behaviours. Finally, our study does not evaluate the effectiveness, appropriateness, or impact of equity- or inclusion-oriented policies or programs; rather, it documents patterns of representation within a citation-defined cohort to inform future hypothesis-driven and longitudinal research.”

d)Regarding the leaking pipeline phenomenon we added the following :”The association between earlier year of first publication and male gender suggests that gender gaps in representation have gradually narrowed over successive academic cohorts. This pattern is consistent with broader European and global trends documented in the She Figures report [9] and the UNESCO Science Report [10], both of which describe steady increases in women's entry into biomedical research over recent decades. Nevertheless, the persistence of a sizable gender gap in the upper tiers of citation impact suggests the presence of the “leaky pipeline” phenomenon, wherein women enter academic medicine in increasing numbers but remain underrepresented in positions of greatest influence [2,16]. Although the cross-sectional design of the present study does not allow direct testing of the leaky pipeline hypothesis, the association between more recent career onset and female gender is consistent with generational shifts described in longitudinal analyses of academic medicine [9,10]. Factors such as differential access to resources, cumulative advantage mechanisms, and citation biases may contribute to these underrepresentations [17].”

Reviewer #3:

Q3 The manuscript is intelligible and formatted in standard English. The data presented are robust, with appropriate statistical treatment. I suggest improving the discussion by including more factors that help to understand this gender disparity, from cultural determinants related to the different social roles assigned to men and women, to issues related to explicit prejudice, harassment, various forms of violence, and implicit biases, which end up distancing women from the hostile environment of academia. Motherhood also has an impact, since it implies great and different responsibilities, and academia should be aware that this impact differs between the male and female genders, resulting in a reduction in the number of female researchers, further diminishing their participation in positions of power, leadership, and decision-making.

A3: We added the following paragraph in the discussion to address Reviewer concern

“Beyond bibliometric and cohort effects, multiple structural and cultural determinants have been shown to influence gender representation in academic medicine. Prior research indicates that women face differential access to leadership opportunities, grant funding, and senior authorship positions, often shaped by implicit bias in evaluation processes and cumulative advantage mechanisms that favor established networks [15,16,18]. Studies examining peer-review and funding decisions suggest that assessments may be influenced not only by the scientific proposal but also by perceptions of the applicant, potentially contributing to slower career progression among women [15]. In addition, qualitative and survey-based research has documented the impact of hostile work environments, harassment, and gender-based discrimination within academic medicine, factors that may reduce retention and advancement of women over time [4,16]. Motherhood and caregiving responsibilities also represent important structural variables: evidence shows that parenthood disproportionately affects women’s research productivity and promotion trajectories, particularly during early and mid-career stages when citation accumulation is most sensitive to output continuity [5,6]. These interacting mechanisms—cultural expectations, implicit bias, structural inequities, and differential caregiving burdens—likely contribute to the persistent underrepresentation of women in positions associated with high citation impact, even in the absence of overt institutional exclusion.

---

## [Decision Letter · Decision Letter 1]

31 Mar 2026

Dear Dr. De Cassai,

Thank you for submitting your manuscript to PLOS ONE. After careful consideration, we feel that it has merit but does not fully meet PLOS ONE’s publication criteria as it currently stands. Therefore, we invite you to submit a revised version of the manuscript that addresses the points raised during the review process.

Specifically, reviewer 2 asked "substantiating evidence for the cultural reference to naming conventions in Italy, normally cultural adduction is not done in scientific writing without empirical justification."

As the corresponding author, your ORCID iD is verified in the submission system and will appear in the published article. PLOS supports the use of ORCID, and we encourage all coauthors to register for an ORCID iD and use it as well. Please encourage your coauthors to verify their ORCID iD within the submission system before final acceptance, as unverified ORCID iDs will not appear in the published article. *Only* the individual author can complete the verification step; PLOS staff the individual author can complete the verification step; PLOS staff the individual author can complete the verification step; PLOS staff the individual author can complete the verification step; PLOS staff *cannot* verify ORCID iDs on behalf of authors.verify ORCID iDs on behalf of authors.verify ORCID iDs on behalf of authors.verify ORCID iDs on behalf of authors.

We look forward to receiving your revised manuscript.

Kind regards,

Claudia Noemi González Brambila, Ph.D.

Academic Editor

PLOS One

Journal Requirements:

Reviewers' comments:

Reviewer's Responses to Questions

**Comments to the Author**

Reviewer #2: All comments have been addressed

Reviewer #3: All comments have been addressed

2. Is the manuscript technically sound, and do the data support the conclusions?

Reviewer #2: Yes

Reviewer #3: Yes

3. Has the statistical analysis been performed appropriately and rigorously?

Reviewer #2: Yes

Reviewer #3: Yes

4. Have the authors made all data underlying the findings in their manuscript fully available?

Reviewer #2: Yes

Reviewer #3: Yes

5. Is the manuscript presented in an intelligible fashion and written in standard English?

Reviewer #2: Yes

Reviewer #3: Yes

Reviewer #2: All Reviewer comments have been successfully addressed, no further changes appear needed except perhaps substantiating evidence for the cultural reference to naming conventions in Italy, normally cultural adduction is not done in scientific writing without empirical justification.

Reviewer #3: Agradeço ao(s) autoro (es) a submissão da versão revisada do manuscrito e por considerar cuidadosamente os comentários levantados durante o processo de revisão.

O(s) autor(es) respondeu de forma satisfatória e construtiva os apontamentos na revisão inicial. As revisões aprimoraram a clareza e a qualidade geral do trabalho. Dessa forma. recomendo a aceitação do seu manuscrito em seu formato atual.

.

---

## [Author Response · Author response to Decision Letter 2]

2 Apr 2026

Padova 01 April 2026

Dear Editor,

we have revised our manuscript according to your comments.

EDITOR COMMENT:

Q1:“Specifically, reviewer 2 asked "substantiating evidence for the cultural reference to naming conventions in Italy, normally cultural adduction is not done in scientific writing without empirical justification."

A1:we have modified the text in order to strength our methods as follows:

“To minimize misclassification bias, we restricted our analysis to Italian scientists, as Italian given names allow gender inference with high accuracy due to morphological regularities, although exceptions prevent deterministic classification. Moreover, the Italian National Institute of Statistics (ISTAT), the official governmental body responsible for collecting and analyzing demographic data in Italy, provides a publicly accessible database (https://www.istat.it/dati/calcolatori/contanomi/; last accessed 01 April 2026) listing Italian given names along with the corresponding percentages of individuals by gender.”

REVIEWERS COMMENTS

Reviewer #2:

R2: All Reviewer comments have been successfully addressed, no further changes appear needed except perhaps substantiating evidence for the cultural reference to naming conventions in Italy, normally cultural adduction is not done in scientific writing without empirical justification.

A2::we have modified the text in order to strength our methods as follows:

“To minimize misclassification bias, we restricted our analysis to Italian scientists, as Italian given names allow gender inference with high accuracy due to morphological regularities, although exceptions prevent deterministic classification. Moreover, the Italian National Institute of Statistics (ISTAT), the official governmental body responsible for collecting and analyzing demographic data in Italy, provides a publicly accessible database (https://www.istat.it/dati/calcolatori/contanomi/; last accessed 01 April 2026) listing Italian given names along with the corresponding percentages of individuals by gender.”

Reviewer #3

R3: Agradeço ao(s) autoro (es) a submissão da versão revisada do manuscrito e por considerar cuidadosamente os comentários levantados durante o processo de revisão.

O(s) autor(es) respondeu de forma satisfatória e construtiva os apontamentos na revisão inicial. As revisões aprimoraram a clareza e a qualidade geral do trabalho. Dessa forma. recomendo a aceitação do seu manuscrito em seu formato atual.

A3: We would like to thank the Reviewer for his help in proving aour manuscript

---

## [Editor Report · Decision Letter 2]

8 Apr 2026

Gender Disparities in Italian Academic Medicine: A Cross-Sectional Study of Clinicians in the 2024 Stanford Top 2% Scientists Database

PONE-D-25-64474R2

Dear Dr. De Cassai,

We’re pleased to inform you that your manuscript has been judged scientifically suitable for publication and will be formally accepted for publication once it meets all outstanding technical requirements.

Kind regards,

Claudia Noemi González Brambila, Ph.D.

Academic Editor

PLOS One
---

## [Editor Report · Acceptance letter]

PONE-D-25-64474R2

PLOS One

Dear Dr. De Cassai,

I'm pleased to inform you that your manuscript has been deemed suitable for publication in PLOS One. Congratulations! Your manuscript is now being handed over to our production team.

Kind regards,

on behalf of

Dr. Claudia Noemi González Brambila

Academic Editor

PLOS One